# Fucoxanthin Production of Microalgae under Different Culture Factors: A Systematic Review

**DOI:** 10.3390/md20100592

**Published:** 2022-09-22

**Authors:** Yam Sim Khaw, Fatimah Md Yusoff, Hui Teng Tan, Nur Amirah Izyan Noor Mazli, Muhammad Farhan Nazarudin, Noor Azmi Shaharuddin, Abdul Rahman Omar, Kazutaka Takahashi

**Affiliations:** 1Laboratory of Aquatic Animal Health and Therapeutics, Institute of Bioscience, Universiti Putra Malaysia, Serdang 43400, Selangor, Malaysia; 2Department of Aquaculture, Faculty of Agriculture, Universiti Putra Malaysia, Serdang 43400, Selangor, Malaysia; 3International Institute of Aquaculture and Aquatic Sciences, Port Dickson 71050, Negeri Sembilan, Malaysia; 4Department of Biochemistry, Faculty of Biotechnology and Biomolecular Sciences, Universiti Putra Malaysia, Serdang 43400, Selangor, Malaysia; 5Laboratory of Vaccines and Immunotherapeutic, Institute of Bioscience, Universiti Putra Malaysia, Serdang 43400, Selangor, Malaysia; 6Department of Aquatic Bioscience, Graduate School of Agricultural and Life Sciences, The University of Tokyo, Bunkyo City, Tokyo 113-8657, Japan

**Keywords:** microalgae, carotenoids, fucoxanthin, systematic review, culture condition, production, macroalgae, biorefinery, molecules

## Abstract

Fucoxanthin is one of the light-harvesting pigments in brown microalgae, which is increasingly gaining attention due to its numerous health-promoting properties. Currently, the production of microalgal fucoxanthin is not yet feasible from an economic perspective. However, the cultivation of microalgae at favourable conditions holds great potential to increase the viability of this fucoxanthin source. Hence, this study aimed to review the fucoxanthin production of microalgae under different conditions systematically. A literature search was performed using the Web of Science, Scopus and PubMed databases. A total of 188 articles were downloaded and 28 articles were selected for the current review by two independent authors. Microalgae appeared to be a more reliable fucoxanthin source compared to macroalgae. Overall, a consensus fucoxanthin production condition was obtained and proposed: light intensity ranging from 10 to 100 µmol/m^2^/s could achieve a higher fucoxanthin content. However, the optimal light condition in producing fucoxanthin is species-specific. The current review serves as an antecedent by offering insights into the fucoxanthin-producing microalgae response to different culture factors via a systematic analysis. With the current findings and recommendations, the feasibility of producing fucoxanthin commercially could be enhanced and possibly achieve practical and sustainable fucoxanthin production.

## 1. Introduction

Fucoxanthin is the light-harvesting pigment associated with photosynthetic energy transfer to chlorophyll *a*. Recently, fucoxanthin has escalated interest due to its numerous biological and health-stimulating properties such as antidiabetic, anti-obesity, anticancer and antioxidant [1,2]. Due to its highly valuable properties, purified fucoxanthin prices range from 40,000 to 80,000 USD/kg, depending on the degree of purity and concentration [3]. Approximately 500 t of fucoxanthin were produced globally in 2016, and an annual increase of 5.3% was estimated between 2016 and 2021 [3].

The fucoxanthin concentration of three diatoms (unicellular microalgae with a silica cell wall), *Thalassiosira weissflogii*, *Phaeodactylum tricornutum* and *Cylindrotheca fusiformis* was elevated during the exponential phase and declined in the stationary phase [4,5]. Inversely, the maximum cell number was observed at the stationary phase [5]. This accentuated the trade-off between growth and fucoxanthin production as one of the primary concerns for developing valuable algal products. In addition to maximising the biomass concentration, sustainable algal biorefinery is also targeted to fully utilise the algal biomass to produce multiple valuable compounds. Thus, the conditions of fucoxanthin-producing microalgae to generate a higher biomass and fucoxanthin are significant.

Several unsystematic narrative reviews have summarised fucoxanthin production from algae [6,7,8,9]. These reviews typically involve only the subjective selection of previous studies by the authors, which could be biased and often lead to a dispute with the available evidence. A systematic review is a review that focuses on a formulated question that uses systematic and explicit methods to identify, select and critically evaluate the relevant primary research and to extract and analyse the data from these studies. A systematic review could minimise the bias efficiently and increase the conclusion validity of the individual studies. However, only a handful of reviews focused on fucoxanthin using a systematic approach [10,11]. Mohibbullah et al. systematically reviewed the pharmacological properties of fucoxanthin and its molecular mechanism for health benefits [10]. In another study, a meta-analysis was performed to examine the effect of the algae subtype and extraction condition on fucoxanthin antioxidant properties [11]. The systematic review on fucoxanthin production by microalgae under different culture conditions is still absent. Hence, this study aimed to examine the effects of several factors, i.e., light, nutrients, salinity, temperature, carbon dioxide and other factors, on the production of biomass concentration and fucoxanthin content. It is particularly essential to understand the favourable conditions under these factors, because it reduces the time and economic cost. At the same time, it also benefits the realisation of commercial fucoxanthin production and maximising the profit. Nevertheless, it is noteworthy that optimal culture conditions in producing both biomass and fucoxanthin are species-specific. The review also discusses the potential future research to fill the knowledge gaps and practical methods to produce fucoxanthin at a large scale with a more cost-effective approach.

## 2. Results

A total of 188 articles were retrieved from three different databases. Most of the articles were in the Web of Science database (*n* = 94), followed by the Scopus database (*n* = 66) and PubMed database (*n* = 28) (Figure 1). After removing the duplicates, 130 articles were obtained. Next, these articles were screened for nonoriginal documents. The remaining articles (*n* = 99) were reviewed based on titles and abstracts to ensure they were related to the topic. After the screening, potentially eligible studies (*n* = 71) were retrieved in full text. They were assessed further for eligibility based on the defined criteria: (1) inclusion of studies, specifically fucoxanthin production of microalgae involving fucoxanthin content or both growth (biomass concentration) and fucoxanthin content under different culture conditions, and the (2) exclusion of studies involving environmental studies (studies utilised fucoxanthin to analyse the microalgae community in a particular habitat). Finally, 28 articles were included in the present systematic review [4,5,12,13,14,15,16,17,18,19,20,21,22,23,24,25,26,27,28,29,30,31,32,33,34,35,36,37]. The information from these articles was extracted and is summarised in Table 1.

## 3. Discussion

### 3.1. Microalgae as Fucoxanthin Source

Algae that produce fucoxanthin could be classified into two major categories: macroalgae (multicellular organisms) and microalgae (unicellular organisms). The commercial-scale production of fucoxanthin relies on the waste part of brown macroalgae [38]. Several studies compared the fucoxanthin content of microalgae and macroalgae [25,39,40]. The fucoxanthin content of the macroalgae ranged from 0.001 to 0.356% DW [16,25]. Microalgae demonstrated that their fucoxanthin content is up to 100 times higher than that of macroalgae. Hence, the production of fucoxanthin from these macroalgae is not commercially feasible. Besides the low fucoxanthin content of these macroalgae, other shortcomings, such as low growth rate and poor product quality, were reported [36]. Overall, microalgae are considered a promising and sustainable source of fucoxanthin due to the high fucoxanthin content and superior growing rates [25,36].

Fucoxanthin production depends on the type of microalgae, growth, fucoxanthin content and extraction efficiency. The fucoxanthin production of a microalga varied depending on the culture conditions. Some microalgae species posed a relatively high percentage of fucoxanthin content (without optimal culture conditions), with a range of 0.22–1.82% of the total dry weight (DW) (Table 2) [39,40,41,42,43,44,45,46,47]. For instance, *Isochrysis galbana* (0.22–1.82% DW), *Chaetoceros gracilis* (0.22% DW), *Nitzschia* sp. (0.49% DW), *Chaetoceros calcitrans* (0.51–1.75% DW), *Cylindrotheca closterium* (0.52%) and *Phaeodactylum tricornutum* (0.86–1.86% DW) (Table 2). A lower percentage of fucoxanthin content was found in other microalgae and even the same species, such as *P. tricornutum* (0.01% DW), *Skeletonema costatum* (0.04% DW), *Odontella sinensis* (0.12% DW) and *Nitzchia laevis* (0.17% DW) (Table 2). It could be due to the utilisation of different cultivation conditions or/and extraction methods in obtaining fucoxanthin. *Isochrysis galbana* (*Tisochrysis lutea* is also known as *I. galbana* T-Iso) and *P. tricornutum* were the most frequent microalgae studied for fucoxanthin production based on Table 1. So far, the highest fucoxanthin content after optimisation of the culture conditions was found in *Tisochrysis lutea* (7.94% DW) (Table 1).

To date, most efforts in studying the fucoxanthin content of microalgae have been conducted with marine species due to their high cellular contents of the compounds of interest, excellent growth rates and the possibility to grow in the open sea ponds in coastal regions. However, culturing marine microalgae in stainless-steel photobioreactors could produce problems associated with corrosion and crystallisation, and an additional step is required to remove the salt from the biomass. To encounter these problems, freshwater microalgae could be used as the surrogate, but the fucoxanthin production from these organisms remains uncharacterised [20]. Freshwater species, particularly *Mallomonas* sp. SBV13 [22], could exhibit substantial growth and fucoxanthin contents (Table 2). Hence, freshwater microalgae deserve to be widely investigated for potential alternative fucoxanthin sources, as they are currently underexplored. 

To sum up, the growth (biomass concentration) and fucoxanthin content are two parameters that are directly responsible for the fucoxanthin production of microalgae. A successful culturing approach for targeted biomass production is essential to ensure fucoxanthin production is sustainable, feasible and economically viable. Cultivating these microalgae at favourable culture conditions could enhance the growth and fucoxanthin content. Integrated processes, including the harvesting and cleaning of microalgal biomass and fucoxanthin extraction, would also need to be assessed to find a robust approach to ensure the quality and sustainability of the fucoxanthin supply.

### 3.2. Effect of Light

The light factor has been frequently examined in the growth and fucoxanthin contents of microalgae. Light is the primary energy source that fuels the biochemical processes in microalgae. The role of light in the growth and fucoxanthin production of microalgae has been investigated. For instance, during the light effect (30 µmol/m^2^/s) on the growth and fucoxanthin production of *Cyclotella cryptica* [34], the light-supplied cells reached the greatest biomass concentration (1.30 g/L), which was at least 20% higher than the culture in dark conditions. Light also enhanced the fucoxanthin accumulation, with a peak content of 0.89% DW. Two important light parameters have been studied in the biomass and fucoxanthin production of microalgae: light intensity and spectra.

Each fucoxanthin-producing microalgae prefers a specific light intensity to achieve the highest growth (biomass concentration) (Table 1). Even within the same genus, they require different light intensities to reach the peak biomass concentration. For example, *T. lutea* (also known as *I. galbana* T-Iso) [32] and *Isochrysis zhangjiangensis* [35] showed the greatest biomass concentrations (1.91 and 2.4 g/L, respectively) at the same light intensity (300 µmol/m^2^/s), whereas the highest biomass concentration (approximately 2.6 g/L) of *Isochrysis* sp. CCMP1324 was observed under a different light intensity, 60 µmol/m^2^/s [31]. This is also applied to the same microalgae species, such as *P. tricornutum*. The maximum biomass concentration (0.29 g/L) of this microalga was obtained under 150 µmol/m^2^/s [33]. In comparison, 70 µmol/m^2^/s was the other light intensity utilised by the same microalga to reach the peak biomass concentration (1.56 g/L) [4]. In the other study, the same microalga achieved the maximum biomass concentration (4.80 g/L) [30]. *Cyclotella cryptica* CCMP333 adopted a low light intensity of 30 µmol/m2/s to attain the greatest biomass concentration (approximately 1.25 g/L) [34]. The possible explanation for these phenomena is prolonged exposure of these fucoxanthin-producing microalgae under a particular culture condition (acclimatisation) before the different light intensity experiments. For instance, *C. cryptica* CCMP333 was cultivated in the dark for 4 days prior to exposure to different light intensities [34] and might require a lower light intensity to achieve the highest biomass concentration. On the other hand, *Nitzschia laevis* reached the greatest biomass concentration (2.22 g/L) without light. This condition could be due to this microalga being cultivated under the heterotrophic mode initially. Therefore, the suitable light intensity to generate the desired biomass concentration could be highly dependent on the previous cultivation conditions of the microalgae.

A conspicuous trend of fucoxanthin accumulation of microalgae was observed (Table 1). A low light intensity fluctuating from 10 to 100 µmol/m^2^/s induced the most fucoxanthin content of microalgae, ranging from 0.52 to 4.28% DW. Higher light intensity led to a decrease in fucoxanthin content. This persistent event may be explained by the fact that fucoxanthin is a light-harvesting pigment, and the elevated level of fucoxanthin compensates for a reduced light intensity [33]. The violaxanthin cycle is a balanced cycle responding to light intensity variation that produces fucoxanthin. A strong irradiance results in the accumulation of zeaxanthin at the expense of violaxanthin and indirectly reduces the level of fucoxanthin. On the other hand, fucoxanthin can also be synthesised via the diadinoxanthin cycle. The high light condition leads to the conversion of diadinoxanthin to diatoxanthin, consequently reducing the biosynthesis of fucoxanthin [31].

Although the adaptability of microalgae to environmental stress, particularly light intensity, is species-specific, there is a similar trend among fucoxanthin-producing microalgae in producing fucoxanthin content. Low light intensity induced a higher accumulation of fucoxanthin. Light intensity ranging from 10 to 100 µmol/m^2^/s is suggested to increase the fucoxanthin content. Nevertheless, the optimal light intensity in producing fucoxanthin is species-specific. On the other hand, the best growth characteristic of fucoxanthin-producing microalgae was obtained at a higher light intensity for photoautotrophic mode. However, the optimal growth of microalgae under the light intensity factor is highly species-dependent. There is a trade between biomass concentration and fucoxanthin production at different light intensities. The possible solution for this issue is discussed in Section 5. Fucoxanthin productivity is recommended as an ideal unit to indicate the fucoxanthin production capability of a microalga, because it considers both biomass concentration and fucoxanthin content simultaneously [35]. However, many studies did not report their fucoxanthin productivity.

Aside from light intensity, the light spectrum is also one of the most pivotal parameters affecting microalgal growth and biochemical composition. The red and blue spectral regions are absorbed by the photosynthetic apparatus of microalgae among a wide region of wavelengths. There is a contradiction between the usage of RL and BL, as both have been reported to boost the growth and fucoxanthin content of microalgae [5,12]. The increase of biomass concentration and fucoxanthin content of *Thalassiosira weissflogii* using BL has been demonstrated [5] (Table 1). Similarly, BL also induced a higher fucoxanthin content of *Nitzschia laevis* [36] (1.20% DW), *P. tricornutum* (1.63% DW) [30] and *Stauroneis* sp. (0.59% DW) [28]. Meanwhile, RL enhanced the biomass concentration and fucoxanthin content of *Odontella aurita* [12]. There was no significant growth and fucoxanthin concentration improvement of these microalgae using WL [5,12,36]. The RL also improved the biomass concentration of *P. tricornutum* [30]. A combination of RL, BL and even GL was adopted to improve biomass concentration and fucoxanthin content of microalgae [12,37]. A 1.46-fold and 1.06-fold increase in biomass concentration and fucoxanthin content were observed using a combination of RL and BL compared to single RL [12]. On the other hand, BL + RL + GL boosted the biomass concentration and fucoxanthin content by 1.52-fold and 1.03-fold, respectively, compared to single BL [37]. Thus, a combination of different appropriate light spectra is suggested to acquire the desired amount of biomass concentration and fucoxanthin content.

### 3.3. Effect of Nutrients

The production of target compounds highly depends on the microalgal culture conditions, especially nutrient supply. Nitrogen is a major macronutrient that is required by microalgae to grow, which is essential in microalgal protein synthesis and accounts for 7–20% of cell DW. The importance of nitrogen sources in producing fucoxanthin has been illustrated. For instance, two phases, nitrogen supply (4 mM) and nitrogen starvation were observed in batch cultivation of *T. lutea* [48]. In the first three days of exponential growth (nitrogen supply phase), the fucoxanthin content increased by 91.66% and decreased by 52.94% after three days (nitrogen starvation) [48].

The influence of different nitrogen concentrations on the biomass concentration and fucoxanthin production of microalgae were examined. Two different nitrogen concentration units were used, millimolar and gram per litre (Table 1). Assuming the sodium nitrate was prepared in 1 L volume, 4 mM is equivalent to 340 mg/L. A high concentration of nitrogen (≥300 mg/L) boosted the growth (either biomass concentration or cell number) of the majority of fucoxanthin-producing microalgae [4,13,14,25,26]. For example, the highest biomass concentration of *P. tricornutum* (1.61 g/L) and *Cylindrotheca fusiformis* (1.52 g/L) was obtained using 300 mg/L sodium nitrate, and these biomass concentrations were at least 3.90-fold higher compared to those without nitrogen supply [4]. However, some microalgae required a lower nitrogen concentration (75–160 mg/L) to achieve the maximum biomass concentration [5,23,31]. No significant improvement in biomass concentration was observed when using higher nitrogen concentrations (150–300 mg/L) [5,31]. Although a low 25 mg/L nitrogen concentration was able to sustain the growth of *Isochrysis* sp. CCMP1324 for the first two days, nitrogen depletion was detected in cultures with 25 and 50 mg/L nitrogen concentration on days 2 and 4, respectively. For a culture with 100 mg/L nitrogen concentration, only a tiny amount of nitrogen concentration was left on day 6 [31]. In contrast, extremely high concentrations of nitrogen (i.e., 9 g/L) could exert stress on the culture by decreasing the pH and consequently inhibit the microalgal growth indirectly [14].

High nitrogen concentrations (≥300 mg/L) favour more fucoxanthin accumulation of the microalgae (ranged from 0.54 to 2.33% DW) [4,5,13,14,25,26]. For example, a 300-mg/L nitrogen concentration promoted a total of 0.70% DW fucoxanthin of *T. weissflogii*, approximately seven-fold higher than cultures without nitrogen [5]. The photosystem was strongly influenced by the nitrogen supply. Fucoxanthin is a core part of the photosystem II associated with chlorophyll. It is susceptible to nitrogen limitation, and its amount may be affected by the nitrogen availability. This fact was well-illustrated by the study by Xia et al. [14], in which the fucoxanthin content of *O. aurita* was analysed when cultivated under three different nitrogen conditions: initial low nitrogen (ILN), initial high nitrogen (IHN) and initial high nitrogen plus supplementary nitrogen (SN). An initial 6 mM and 18 mM nitrogen concentration were applied in ILN and IHN, respectively. For SN, a total of 18 mM nitrogen concentration was supplied at the beginning, and an additional 6 mM nitrogen concentration was added on day 3, day 7 and day 11 (36 mM in total). A declining trend was identified for the fucoxanthin content of this microalga in the ILN group due to nitrogen depletion. Due to the constant nitrogen supply, the fucoxanthin content of this microalga in the SN group was gradually improved to the highest amount, 2.33% DW. To recapitulate, more studies involving a wide range of fucoxanthin-producing microalgae are needed to investigate the amount of nitrogen that benefits the growth and fucoxanthin content of microalgae for sustainable cultivation.

Different nitrogen sources on the growth and fucoxanthin production of microalgae were also investigated. For instance, alternative nitrogen sources such as urea, sewage water and liquid fertiliser were assessed on the fucoxanthin production of *C. gracilis* [18]. The authors reported that fucoxanthin production using urea (1.95 mg/L) was similar to nitrate (2.01 mg/L). A lower fucoxanthin yield was observed using 2% liquid fertiliser (1.39 mg/L) and 67% sewage water (0.43 mg/L). Meanwhile, four different nitrogen sources (200 mg/L) such as sodium nitrate (NaNO_3_), potassium nitrate (KNO_3_), urea (CO(NH_2_)_2_) and ammonium chloride (NH_4_Cl) in the Daigo’s IMK medium were tested on the biomass concentration and fucoxanthin production of *Pavlova* sp. OPMS 30543 [16]. Media consisting of KNO_3_ boosted the highest biomass concentration (1.8 g/L), while the lowest biomass concentration (0.82 g/L) was obtained using NH_4_Cl as a nitrogen source. The media with NaNO_3_ resulted in a greater fucoxanthin content (12.74 mg/g DW), followed by CO(NH_2_)_2_ (8.38 mg/g DW), NH_4_Cl (7.80 mg/g DW) and KNO_3_ (5.57 mg/g DW). The study concluded that the media with NaNO_3_ yielded the highest fucoxanthin production. Alternatively, a series of nitrogen sources (bacterial growth media) such as yeast extract (YE), tryptone (Try) and a combination of YE and Try were utilised to enrich the f/2 medium in cultivating *P. tricornutum* [19]. These bacterial growth media boosted the biomass concentration with the highest (approximately 0.24 g/L) using the combination of YE and Try, which was 3.48-fold higher than the non-enriched f/2 medium. Similarly, the combination of YE and Try produced the maximum fucoxanthin yield (around 7.5 mg/L). Although urea showed a comparative fucoxanthin yield with commercial nitrate and is a cheaper alternative nitrogen source, the adverse effects of urea on microalgae were illustrated. Hence, commercial nitrate such as NaNO_3_ is currently suggested for microalgal cultivation, despite the high demand for a cheaper surrogate nitrogen source. Fortifying the conventional algal culture media such as f/2 medium using bacterial growth media opens a new horizon of alternative cost-effective approaches to improve the microalgal growth and fucoxanthin content. Additional studies are needed to explore the possibility of these bacterial growth media in enhancing the growth and fucoxanthin content of the other fucoxanthin-producing microalgae.

Phosphorus plays a key role in the normal growth and development of algal cells. Compared to nitrogen, phosphorus is the main limiting nutrient for microalgae in the environment. Phosphorus only constitutes around 1% of the cell DW. A high phosphate concentration (>4.5 mg/L) could stimulate the peak growth and fucoxanthin content of microalgae. For example, Sun et al. utilised four phosphorus concentrations (1.13, 2.25 and 4.5 mg/L) to explore the effect on *Isochrysis* sp. CCMP1324 [31] (Table 1). The 4.5 mg/L phosphate induced the highest biomass (around 2.75 mg/L) and the maximum fucoxanthin content (around 13.5 mg/g DW) of this microalga. Yet, only a small portion of phosphorus was left in the medium after the six-day cultivation. The other study examined the effect of five different phosphate concentrations (0, 2.5, 5, 10 and 20 mg/L) on the growth and fucoxanthin production of *T. weissflogii* [5]. The greatest cell number (3.6 × 10^6^ cells/mL) and fucoxanthin content (around 0.65% DW) were obtained at phosphate concentration of 10 mg/L. Nevertheless, there was no significant difference between 10 mg/L and 20 mg/L phosphate for both the cell number and fucoxanthin content. The impact of phosphorus concentration was less conspicuous than nitrogen concentration on the growth and fucoxanthin content of microalgae [31].

Silicate is an essential nutrient for diatom growth, as silicate is needed to build the rigid external frustule shells of diatoms. Cell growth and morphology will be interrupted by the silicate deficiency condition. Marella and Tiwari et al. [5] studied the effect of five different silicate concentrations (0, 35, 70, 140 and 280 mg/L) on the growth and fucoxanthin production of *T. weissflogii*. An increased silica concentration yielded an elevation trend of the cell number and fucoxanthin content. The optimum silica concentration to attain the highest cell number (around 3.9 × 10^6^ cells/mL) and fucoxanthin content (around 0.75% DW) was the same, which was 280 mg/L silicate. High silica concentrations (>280 mg/L) could result in higher growth and fucoxanthin content of microalgae. However, these properties are likely to depend on light intensities in addition to silicate concentrations. Different concentrations of silicate and various photon fluxes showed diverse impacts on the fucoxanthin content of *P. tricornutum* [27]. Under 255 µmol/m^2^/s light intensity, *P. tricornutum* accumulated more fucoxanthin at 3 mM silicate medium (or equivalent to 366.18 g/L, assuming sodium silicate was prepared in a 1-L volume) compared to a 0.3 mM silicate medium. Meanwhile, the fucoxanthin content of this microalga under 204 µmol/m^2^/s was reduced in a 3.0 mM silicate medium compared to that in 0.3 mM silicate medium. The other study investigated the cellular response of *Navicula laevis* to different silicate concentrations (0, 64, 240, 480 and 960 mg/L) under the heterotrophic cultivation mode [15]. The medium without silicate produced the smallest amount of biomass (less than 0.5 g/L) of this microalga. Conversely, the highest biomass (2.41 g/L) was acquired using the medium with 480 mg/L silicate on day 4. The fucoxanthin content was extensively boosted with the high silicate concentrations (240, 480 and 960 mg/L), while without or a low silicate concentration hindered the fucoxanthin accumulation. Similar to biomass, the peak fucoxanthin yield (32.78 mg/L or fucoxanthin content: 1.22% DW) was also attained at 480 mg/L silicate [15]. Parkes et al. illustrated the importance of the nutrients (f/2, f/2 without silicate, f/2 without nitrate and normal seawater) on the fucoxanthin content of *Stauroneis* sp. The f/2 medium with complete nutrients (silicate and nitrate) resulted in the greatest fucoxanthin content (0.59% DW) in the study. 

Carbon is a prominent nutrient that controls the growth and metabolism of microalgae. The impact of additional different carbon sources (glucose, methanol, sodium acetate and sodium bicarbonate) to the Daigo’s IMK medium on the biomass concentration, fucoxanthin content and yield of *Pavlova* sp. OPMS 30543 was investigated [16]. These supplementary carbon sources improved the biomass concentration compared to the medium without the source. The highest biomass concentration (1.79 g/L) was attained with the addition of sodium acetate, followed by sodium bicarbonate (1.28 g/L), glucose (1.19 g/L) and methanol (0.71 g/L). This microalga grown in the medium with methanol exhibited higher fucoxanthin content (7.26 mg/g DW) compared to the medium consisting of glucose (4.25 mg/g DW), sodium acetate (4.11 mg/g DW) and sodium bicarbonate (2.99 mg/g DW). The best fucoxanthin yield (7.36 mg/L) was obtained when sodium acetate was supplemented to the medium. More studies are required to determine the suitable amount and type of carbon source for improving the growth and fucoxanthin content of microalgae.

### 3.4. Effect of Culture Media

Culture media is composed of specific mixtures of nutrients that sustain high microalgal biomass accumulation and metabolite productivity. Most studies focused on seawater-based culture media in inducing efficient biomass and fucoxanthin production. For instance, Kanamoto et al. compared 50% seawater enriched with either 2X Daigo IMK, f/2 or Walne media in producing the greatest biomass concentration and fucoxanthin content of *Pavlova* sp. OPMS 30543 [16]. The cultivation of this microalga in 2X Daigo IMK medium resulted in the highest biomass concentration (0.92 g/L) and fucoxanthin content (0.26% DW) among the other culture media. Moreover, the other study analysed the growth and fucoxanthin accumulation of *Chaetoceros grarcilis* when cultivated in two different culture media, f/2 medium and the Daigo IMK medium [18]. A higher cell density (5.22 × 10^6^ cells/mL) and fucoxanthin content (2.02 mg/L) were obtained using the Daigo IMK medium than the f/2 medium. It could be due to the major difference in the nitrate concentration of these two media, the Daigo IMK (200 mg/L) and f/2 media (75 mg/L). On the other hand, different f/2 medium compositions (f/2, 10 × f/2 and f/2 + 10 × nitrate) on the growth and fucoxanthin content of *P. tricornutum* were studied [33]. The utilisation of 10x f/2 medium generated the greatest biomass concentration (0.59 g/L) of this microalga, whereas f/2 medium with 10× nitrate supplementation promoted the maximum fucoxanthin content (5.92% DW). This indicated that algal growth was restricted by factors other than nitrogen and fucoxanthin production likely depends on nitrogen availability in the medium. Meanwhile, Butler et al. [17] investigated a number of commercial powdered media (Cell-Hi F2P, JWP and WP) and a hydroponics medium (FloraMicroBloom) in culturing *P. tricornutum* and then compared it with an f/2 medium. The Cell-Hi JWP medium produced the maximum biomass concentration (0.45 g/L) and fucoxanthin content (1.33% DW) among the tested media. Additionally, Cell-Hi JWP medium (EUR 6.03 per kg dry biomass) was reported to be cheaper than f/2 medium (EUR 16.88 per kg dry biomass). The Daigo IMK medium seems a promising culture medium to acquire the desired growth and fucoxanthin production of marine microalgae. However, it appears that comparison of the 2× Daigo IMK medium with the f/2 and the Walne media is inappropriate as the 2× Daigo IMK medium consisted of two times more nutrients than the other media [16]. More studies are required to compare the Daigo IMK medium with the other marine microalgal culture media in improving the growth and fucoxanthin production of marine microalgae.

A freshwater-based culture media, Freshwater Diatom Media (FDMed) was developed by Gerin et al. to improve the growth and fucoxanthin content of two freshwater diatom species (*Sellaphora minima* and *Nitzchia palea*) compared to Guillard and Lorenzen’s WC medium and the Modified COMBO medium [20]. The biomass concentration of *S. minima* and *N. palea* was 1.71 g/L and 1.19 g/L, respectively, using the FDMed and was much higher than those using the other media. Similarly, the fucoxanthin content of *S. minima* (0.75% DW) and *N. palea* (0.55% DW) cultivated in FDMed were greater than those using the other media. Only one study compared the freshwater culture media to boost growth and fucoxanthin production of microalgae. The designed freshwater media, FDMed, surpassed the conventional culture media in promoting the growth and fucoxanthin production of microalgae [20]. Similar to seawater-based culture media, more studies are required to verify the capability of this culture media.

### 3.5. Effect of Salinity

Salinity is one of the vital factors that can regulate the growth rate and biochemical composition of microalgae, mainly marine and brackish microalgae. Optimum salinity is highly dependent on the microalgal species. For instance, the optimum salinity level for *Chaetoceros muelleri* (marine species) to reach its maximum fucoxanthin content was 45‰. At the same time, 85‰ was the optimum salinity level of *Amphora* sp. (halotolerant species) to achieve its greatest fucoxanthin content [29]. Even for the marine species, these microalgae also preferred a specific salinity to strive for growth and produce a higher amount of fucoxanthin. *Isochrysis galbana*, a marine species was exposed to two different salinities, 20‰ and 35‰. This microalga showed a greater growth (around 5.9 × 10^6^ cells/mL) and fucoxanthin content (1.2% DW) under a salinity of 35‰ than 20‰ [13]. Meanwhile, the growth (1.49 g/L) and fucoxanthin content (0.74% DW) of *P. tricornutum* were in favour of a 20‰ salinity level after this microalga was cultivated under four different salinities (5, 10, 20 and 30‰) [4]. The other microalga, *C. fusiformis*, grew better under 30‰ by producing the highest biomass concentration (1.64 g/L), whereas the greatest fucoxanthin content (0.58%) was generated under a salinity of 10‰ [4]. These studies highlighted the obligation of optimising the salinity level for the interest microalgae to yield the peak biomass concentration and fucoxanthin content.

### 3.6. Effect of Temperature

Temperature is considered one of the most important factors influencing the development of microalgae with respect to the growth rate, cell size, biochemical composition and nutrient requirements. Like salinity, each fucoxanthin-producing microalga has its optimum temperature to yield the maximum growth and fucoxanthin content. An optimisation study employing response surface methodology has shown that the optimum temperature for *T. lutea* to generate the greatest fucoxanthin content was 25 °C [24]. In the other study, the same microalgae, *T. lutea*, leaned towards the temperature of 30 °C among the other temperatures (16.5, 20 and 25 °C), with which the peak biomass concentration (1.81 g/L) and fucoxanthin content (around 0.21% DW) were obtained [32]. In addition, a larger cellular diameter was observed at a lower temperature (16.5 °C) compared to a higher temperature (30 °C). The bigger cell size was to adapt at low temperatures due to the accumulation of intracellular lipids. Thus, the optimum temperature of the target microalgae to achieve desirable growth and fucoxanthin content must be determined for fucoxanthin commercialisation.

### 3.7. Effect of Carbon Dioxide

Photoautotrophic microalgae possess the ability to fix carbon dioxide to perform photosynthesis. Therefore, the carbon dioxide level plays a crucial role in microalgal growth and fucoxanthin accumulation. McClure et al. [33] studied the impact of three different carbon dioxide concentrations (0, 1 and 2%) on the growth and fucoxanthin production of *P. tricornutum*. The biomass concentration of this microalga decreased when the carbon dioxide concentration was increased. For example, 2% carbon dioxide concentration had a detrimental effect on the cell concentration, in which the biomass concentration decreased by 62.22% compared to those cultured without carbon dioxide. The highest biomass concentration (0.45 g/L) was obtained without carbon dioxide. The carbon dioxide could lead to the acidification of the media, which reduced the media pH to 6.9 after adding 2% carbon dioxide. In contrast, 1% and 2% carbon dioxide concentrations induced higher fucoxanthin accumulation than 0% carbon dioxide, suggesting either carbon dioxide might be required to produce fucoxanthin or carbon dioxide concentration could be a stress to this microalga and the increase of fucoxanthin to mitigate this stress. The other study examined the supplementation of three different carbon dioxide concentrations (0, 2 and 5%) on the cell growth and fucoxanthin accumulation of *I. zhangjiangensis* [35]. Both 2% and 5% supplementation boosted the biomass concentration and fucoxanthin content of this microalga compared to 0% supplementation. The peak biomass concentration (1.35 g/L) and fucoxanthin content (2.32% DW) were obtained at a 5% carbon dioxide concentration. The possible explanation for the difference between these two studies is the tolerance of microalgae to the acidic pH after adding carbon dioxide. *Isochrysis zhangjiangensis* could be a microalga capable of withstanding more acidic conditions. Hence it could grow well under 5% carbon dioxide concentration. Therefore, more studies are required to verify the demand for carbon dioxide by the microalgae in producing fucoxanthin.

### 3.8. Effect of Oxidative Stress

Microalgae can counteract the detrimental effect of oxidative stress, such as reactive oxygen species, due to the antioxidant activity of carotenoids. It has been reported that an increase in oxidative stress could result in the elevation of carotenoids amount. A set of oxidative stress was studied on the fucoxanthin production of *Amphora capitellata* [21]. Four different oxidative stresses were produced as the following: 0.1 mM hydrogen peroxide + 0.1 mM Fe^2+^, 0.1 mM sodium hypochlorite + 0.1 mM Fe^2+^ and 0.1 mM hydrogen peroxide + 0.1 mM sodium hypochlorite. A combination of 0.1 mM hydrogen peroxide + 0.1 mM sodium hypochlorite improved the biomass concentration (around 0.64 g/L) and fucoxanthin content (4.18% DW) of *A. capitellata* by the most compared to the other oxidative stresses. Singlet molecular oxygen was produced from the aforementioned oxidative stress. Fucoxanthin is a polar carotenoid that blocks this molecular oxygen from penetrating the hydrophobic core of the membrane and protects the cellular components. Thus, the amount of fucoxanthin increased dramatically, and the cellular processes could function normally, resulting in a higher biomass concentration.

## 4. Potential Applications of Fucoxanthin and Current Fucoxanthin-Based Products

Brown seaweeds such as *Saccharina* sp., *Fucus* sp., *Sargassum* sp., *Hijka fusiformis* and *Undaria pinnatifida* are the commercial sources of fucoxanthin for food [49]. They are usually sold in powder or oil in capsules or cachets. They are distributed by several major Asia companies, such as Yangling Ciyuan Biotech Co. and Agrochemi Co., and are authorised by the Novel Food Regulation in Europe.

On the other hand, fucoxanthin from microalgae is utilised as supplementary and whole feed or food, as it could fortify the functional and nutritional value of feed or food. For instance, incorporating fucoxanthin into the feed of broiler chicken improved meat colour and plasma antioxidative status [44] and the egg yolks’ carotenoid content [50]. Meanwhile, the combination of fucoxanthin with whole and skimmed milk demonstrated excellent plasma absorption and organ accumulation rates for fucoxanthin in both in vivo and in vitro studies [51,52]. In addition, the nutritional value of wheat-based pasta was boosted with the addition of whole edible brown seaweed (rich in fucoxanthin).

The several potential applications of fucoxanthin in the cosmetics region have been investigated. A significant decrease in wrinkles and elevation of skin moisture and elasticity in human trials was observed with a wrinkle care cream consisting of 0.03% fucoxanthin [53]. Moreover, Rodriguez-Luna et al. [54] developed a fucoxanthin-containing cream and demonstrated that this cream could prevent exacerbations associated with inflammatory skin diseases and protect skin against UV radiation. Furthermore, the amalgamation of fucoxanthin in solid lipid nanoparticle systems exhibited an excellent sunscreen-boosting effect [55].

So far, several fucoxanthin-based products such as Xanthigen, FucoVital and BrainPhyt^TM^ are available to the public. Xanthigen is an anti-obesity fucoxanthin-based product (https://nektium.com/branded-ingredient/xanthigen/, accessed on 8 June 2022), which has been proven to reduce the body and liver fat content, as well as enhance liver function in human trials [56,57]. On the other hand, FucoVital is the first fucoxanthin-based product that obtained approval from the United States Food and Drug Administration (NDI1048, 2017) (Algatechnologies Ltd., Kibbutz Ketura, Israel). FucoVital is the only microalgae-derived fucoxanthin-based product that can maintain liver health (https://www.algatech.com/algatech-product/fucovital/, accessed on 8 June 2022). BrainPhyt^TM^ is a natural extract (2% fucoxanthin with other extracts) produced by Microphyt Company in Baillargues, France. It is a commercial product that provides a full spectrum of cognitive support, such as enhancing cognitive performance, short- and long-term memory, etc. (https://www.brainphyt.com/, accessed on 8 June 2022). The other fucoxanthin-based product, such as NutriXanthin^TM^ and DermaXanthin^TM^ are produced by AlgaHealth Company (Ein Shemer, Israel) that could be utilised as ingredients for food supplements and cosmetics products (https://alga-health.com/products/, accessed on 8 June 2022).

## 5. Obstacles and Possible Solutions

Microalgae appeared to be a promising source of fucoxanthin. Hence, a large number of research and development programs have been conducted worldwide. However, commercial production of microalgal-based fucoxanthin is far from economically feasible. Numerous obstacles hinder the fucoxanthin production of microalgae (Figure 2). One of them is biological contamination. Different types of biological contaminants could inhibit or devastate the cellular biomass accumulation of the microalgae. At the same time, they also decreased the growth of microalgae indirectly, resulting in a reduction of fucoxanthin productivity. These contaminants are fungi, bacteria, protozoa, rotifer, viruses or undesired alga [58]. For instance, contamination with fungi under the class *Oomycota* could lead to a loss of 10–60% marine microalgae culture [59]. In addition, contamination elevated the cost of microalgal cultivation and resulted in economically infeasible. Biological contamination of *Spirulina platensis* culture by rotifers led to a loss of USD 1230 worth of dry biomass [60]. To date, only one study reported the effect of biological contaminants, protozoa, on the growth and fucoxanthin content of *P. tricornutum* [61]. *Heterolobosean amoeba* was the most destructive protozoan, which can engulf up to 60 *P. tricornutum* cells within 10 min. After being contaminated with this protozoan, a total reduction of 84.62% and 96.80% in biomass productivity and fucoxanthin content of *P. tricornutum* were reported. To overcome the contamination problems, culture monitoring and detection of contaminants are indispensable in the early stage of this issue. Conventional morphological observation approaches such as staining and microscopy are employed to detect and identify the contaminant. After knowing the contaminant, a reliable strategy could be developed to control and even prevent the transmission of contaminants [62]. For example, periodic culture mixing with the usage of ammonium bicarbonate alleviated the damage caused by the amoeba by creating an unfavourable condition for the amoeba and subsequently increased the microalgal biomass concentration [61].

Outdoor cultivation, such as open ponds, is widely utilised as a sustainable cultivation system to produce commercially microalgal biomass and interest compounds [57]. To reduce the reliance on a limited freshwater resource, the open ponds often employed seawater for microalgal cultivation. However, the increase of salinity over time due to evaporation from the pond could greatly reduce the growth and production of interest compounds beyond the optimal point. Occasionally, this might also cease the microalgal development and their activities. Hence, this will lead to unsustainable and economically unfeasible microalgal cultivation. Maintaining high microalgal growth is paramount, as a linear relationship between biomass and fucoxanthin content was reported [29]. A stepwise cultivation approach offers a viable solution to continuously produce biomass and fucoxanthin without the interruption of increased salinity levels. This approach initially grows marine species, followed by halotolerant species, which could produce high biomass and fucoxanthin content and withstand a salinity range of 35–125‰. In addition, the cultivation of the second species in the spent culture media of the first species will minimise the loss of nutrients and the usage of fresh media, subsequently reducing the cost of microalgal cultivation [63]. Hence, the stepwise cultivation approach is a sustainable, practical and potentially cost-effective method for fucoxanthin production.

The climatic feature is an essential element for successful fucoxanthin production. In temperate countries, outdoor microalgal cultures will encounter temperature fluctuations between 10 °C and 45 °C [64]. Some temperatures within that range are above the survival thresholds of most commercialised microalgal species [65]. Temperatures between 15 °C and 30 °C are the temperature that most microalgal species can grow, with the optimal being between 20 °C and 25 °C. Most microalgae cannot grow above 30 °C, whereas the growth and activities of microalgae are affected by the temperature falling below 20 °C during the winter [66]. Therefore, a temperature control system is obligated for microalgal cultivation. Temperature control constituted the second-largest annual energy requirement for microalgal cultivation and product processing [67], which could compromise the economical viable production of fucoxanthin. Moreover, the narrow range of temperature that microalgae can strive for growth limits the production location. A ‘cold-adapted’ *T. lutea* was developed via a continuous low-temperature adaptation experiment to encounter the temperature in the winter season [68]. This strain could grow at 15 °C and the biomass and fucoxanthin content were comparable to the wild *T. lutea* that grew originally at 30 °C. This strain could be cultivated continuously in the winter for fucoxanthin production purposes. Additionally, this strain could be utilised in food or feed industries as the produced strains are non-genetically modified organisms. 

As mentioned earlier, there is a trade-off between growth and fucoxanthin content. Microalgae accumulate more fucoxanthin under low light intensity [22,31], which hurts the growth. Meanwhile, the higher growth of microalgae was observed under a high light intensity [33,35], but this light intensity attenuated the fucoxanthin content. Sustainable fucoxanthin production maximises biomass concentration and fucoxanthin content [5]. A two-stage/phase/step cultivation approach was proposed to address this impediment. Currently, there are three types of this approach. First, a fed-batch *Nitzschia laevis* culture under heterotrophic mode in a 3 L fermenter was established initially and achieved the greatest biomass concentration of 17.25 g/L on day 10 [36]. Next, the culture was subjected to a mixed light of BL and WL (1:1) for two days to induce fucoxanthin accumulation. A total of 1.20% DW fucoxanthin was generated. This resulted in the final fucoxanthin productivity of 16.5 mg/L/day. However, this approach is only applicable to the microalgae that can be cultivated under heterotrophic mode. The second approach utilised a high light intensity (120 µmol/m^2^/s) to increase the biomass production of *P. tricornutum* and *C. fusiformis* in the first step, and then, the accumulation of fucoxanthin was conducted using a low light intensity (30 µmol/m^2^/s) for last three days in the second step [4]. The maximum biomass concentration of *P. tricornutum* and *C. fusiformis* was 1.3 g/L and 1.14 g/L, respectively, on day 18. The highest fucoxanthin content of *P. tricornutum* (0.77% DW) and *C. fusiformis* (0.58% DW) was observed on day 6. The low light intensity on the last three days improved the fucoxanthin accumulation in these microalgae; however, the amount was still lower than the amount of fucoxanthin that was produced on day 6. Therefore, a longer period under low light intensity is suggested to induce the accumulation of fucoxanthin substantially. In the third approach, *P. tricornutum* culture was fed with glycerol and cultivated at RL:BL (6:1) (mixotrophic mode) for six days in phase 1, followed by the light was shifted to RL:BL (5:1) with the addition of tryptone for another six days in phase 2 [30]. The maximum biomass concentration and fucoxanthin content was 6.52 g/L and 1.33% DW, respectively. A total of 8.22 mg/L/day fucoxanthin productivity was obtained using this two-phase cultivation approach. Although this approach showed excellent biomass concentration and fucoxanthin content compared to the second approach, the fucoxanthin productivity was almost two-fold lower than the first approach.

The downstream process, such as the extraction of fucoxanthin, encountered several challenges. Conventional extractions using solvents such as tetrahydrofuran, methanol or dichloromethane are the most described solvent in previous studies [9,69]. However, the usage of these conventional solvents was prohibited due to their toxicity and refute with the environmental regulations. In addition, recovering these solvents using evaporation or distillation is very costly. Moreover, the energy-intensive extraction processes impede the commercialisation of fucoxanthin. Furthermore, the increasing global awareness of sustainability encourages the utilisation of alternative extraction approaches that are green and environmentally friendly [70]. One of these approaches is supercritical CO_2_ extraction, an attractive, eco-friendly and sustainable method to extract fucoxanthin. In contrast, expensive equipment is needed to bring CO_2_ to a supercritical state. Similarly, enzyme-assisted extraction is another green fucoxanthin extraction approach that is time-consuming and very expensive due to the usage of enzymes. In order to achieve a robust fucoxanthin extraction, the features of extraction approach are required to be (1) less solvent consuming, (2) cost effective (3), less energy-consuming and (4) high efficiency. The single-step extraction approaches like liquid biphasic flotation could be the ideal fucoxanthin extraction method. This liquid biphasic flotation approach is a recent technological innovation that combined the mechanism of liquid biphasic and solvent sublation. This approach showed high separation efficiency, ease of operation, scalable, energy efficiency and environmental friendliness [71].

The product safety of fucoxanthin is always a major concern for consumers. Many contaminants such as heavy metals, pesticides and other chemical contaminants could exist in the microalgae. Heavy metals are released into the microalgae via cement, iron and copper tubing and plastics utilised in the production plants and processing equipment [72]. On the other hand, misbranded and counterfeit fucoxanthin-containing dietary supplements in the market are one of the concerns by consumers. A study by Hossain et al. [73] demonstrated that eight out of ten chosen online-sourced fucoxanthin-containing dietary supplements did not meet their label claim. To solve this issue, quality control is essential to warrant the safety, quality and efficacy of the fucoxanthin product. Hence, the authority should monitor and regulate these issues vigilantly according to the World Health Organization and the United States Food and Drug Administration.

## 6. Knowledge Gap and Future Directions

The current review underlined several knowledge gaps that are worthy of further research endeavours (Figure 3). Firstly, most fucoxanthin production studies pay attention to marine species (Table 1). To date, only a handful of studies on fucoxanthin production of freshwater microalgae have been reported [20,22]. Based on Table 1, freshwater microalgae, especially *Mallomonas* sp. SBV13, showed superior biomass concentration and fucoxanthin content compared to most marine species. The major drawback of using freshwater microalgae for fucoxanthin production is that they require a higher water footprint than marine microalgae [74]. Ozkan et al. reported that freshwater microalgae required a total of 2857-L and 1618-L water footprint to produce 1 kg biomass in open raceway ponds and biofilm photobioreactors, respectively [75]. Freshwater availability and quality is the primary concern in many parts of the world. In 2015, almost one-third of the world encountered a shortage of freshwater provision [76]. Nevertheless, using wastewater, seawater or water recycling could lower the water footprint. Recycling the growth media demonstrates that the water footprint of microalgae can be decreased by 90% [77]. It also complies with the circular economy, which recycles materials maintained in a close loop and minimises waste output. Therefore, freshwater microalgae could be one of the promising fucoxanthin sources as they are currently underexplored.

The effect of the different factors on fucoxanthin production by microalgae was described in several previous studies and discussed in the earlier sections. Further, these impact factors on commercial-scale fucoxanthin production are limited. Despite the recent improvements and developments in fucoxanthin production, many of the available studies on the production of fucoxanthin are limited to the laboratory scale, which might impede the scalability of the production. The optimal fucoxanthin production information is particularly crucial at the commercial scale, because there is always a concern that the plunged dry biomass in microalgal cultivation when shifted from the laboratory to pilot scale [78]. For example, microalgal cultivation from the laboratory to pilot scale reduced 22.9–82.0% of the biomass [12,78,79]. It is believed that the decrease of the biomass might also occur from the pilot to industrial scale, which hindered the feasibility of microalgae fucoxanthin production. However, commercial fucoxanthin production could benefit from the optimised parameters based on the laboratory or pilot scale as they might be different only in some key processes or conditions. Obtaining the optimum conditions at the industrial scale will alleviate the economic cost and make fucoxanthin production more economically sustainable.

Producing large amounts of fucoxanthin involves enormous volumes of microalgal culture. Cultivation in an open system is an alternative approach to produce fucoxanthin at a reasonable cost. Nevertheless, this system is prone to contamination. These contaminants could impair the development of the microalgal cultures, subsequently decreasing the interest in compound production and occasionally leading to the collapse of microalgal culture [58]. Furthermore, contamination will elevate the production cost and possibly degrade the quality of the target product [80]. Up to now, most microalgae studies have reported the contamination in green microalgae [81,82] and cyanobacteria [83,84] and rarely in fucoxanthin-producing microalgae. In addition, the impact of these biological contaminants was only restricted to the culture stability and biomass productivity of these microalgae. So far, solely one study has examined the potential influence of predatory protozoa on the biomass and fucoxanthin production of *P. tricornutum* [61]. An analysis of these biological contaminants in fucoxanthin-producing microalgae undoubtedly could contribute to constructing a comprehensive and effective strategy to prevent, eradicate or even mitigate the detrimental effect of these contaminants. All this evidence lends credence to the need for research into the contamination problem in fucoxanthin-producing microalgae, starting with investigative work on the types of contaminants present in the cultures and emphasising the alteration of growth and fucoxanthin content during contamination.

The high cost of microalgae biomass production and the inefficient fucoxanthin extraction currently resulting in microalgae fucoxanthin are still not commercially attractive. High energy-intensive processes such as microalgal cultivation, harvesting and drying restrain the possibility of commercial production of fucoxanthin. A substantial technological advancement emphasises the increment of the target compound, cost reduction and higher biomass productivity needed to improve the feasibility of fucoxanthin production [85]. The concept of a technoeconomic analysis is defined by the terms economic and technology, which are employed to examine the technical feasibility of the commercial approach. In other words, the technoeconomic assessment is an important paramount practice to evaluate the capital and operating costs and risk related to the production process (biorefinery) and technologies that determine the economic feasibility of microalgal target compound development. Moreover, it also helps govern and detect the potential investment and finance processes of a future industry [86]. The economic data on the production cost of fucoxanthin-producing microalgal biomass in photobioreactors were scarce. A study by Derwenskus et al. was the solely available technoeconomic study that reported the economic assessment of a holistic process for the coproduction of fucoxanthin and eicosapentaenoic acid (EPA) using *P. tricornutum* (a fucoxanthin-producing microalga) in flat-panel airlift photobioreactors with artificial lighting [67]. The economic cost of three production scenarios: the basic scenario, the industrial scenario and the hypothetical industrial scenario, was calculated based on experimental data. The substantial reduction of the biomass cost (392-228 EUR) through an increase in the production capacity from the basic scenario (0.7 t/year) to the hypothetical industrial scenario (170 t/year) resulted in a production cost of 7343 EUR/kg for concentrated EPA and 32,042 EUR/kg for fucoxanthin (purity > 90%). The technoeconomic analysis of microalgal fucoxanthin production is quite variable, as it relies on several factors such as the scale of a production facility, type and design of the cultivation system and upstream and downstream processes [86]. Therefore, there is a dire need to uncover the fucoxanthin production cost closer to the real world. This information could bring operational practicability and profitability in microalgae to fucoxanthin production in a more realistic approach.

## 7. Materials and Methods

### 7.1. Literature Search Strategy

In this systematic review, three different search engines: Scopus, Web of Science and PubMed databases, were utilised to acquire the scientific articles (collected on 30 February 2022). The keywords of “fucoxanthin” AND “fucoxanthin production” OR “alga*” AND “microalga*” were inserted into the databases to search for the literature. There was no restricted time frame for the retrieved articles.

### 7.2. Screening and Eligibility Criteria

In the screening, the Preferred Reporting Items for Systematic Review Meta-Analysis (PRISMA) guidelines were followed to filter the searched articles. The inclusion criteria were: (1) full-text research articles and (2) studies related to fucoxanthin production of microalgae concerning culture conditions to enhance the production. On the other hand, the exclusion criteria were: (1) irrelevant papers; (2) languages other than English; (3) inaccessible full-text and (4) non-original research documents such as monographs, theses, dissertations, review articles, letters, book chapters, conference abstracts, reports, proceeding papers and patents.

### 7.3. Data Extraction

A review team composed of two independent authors was established to minimise the random errors and bias at all review stages. Firstly, duplicated articles from these three databases were omitted from this review. Subsequently, the nonoriginal research documents were discarded. The titles and abstracts of the articles were screened against the eligibility criteria. Next, the full texts of the remaining articles were evaluated to check their eligibility. Discrepancies on whether a given article should be included or excluded were resolved through discussion until a consensus was achieved. The data of the selected articles were extracted and entered into an Excel file. The Excel data contained information such as the name of microalgae, culture condition, biomass concentration and fucoxanthin content. 

## 8. Conclusions

Microalgae are on the cusp of a new era, and the current work has revealed the pertinent knowledge gap of culture conditions to achieve practical and sustainable commercial fucoxanthin production. Recently, fucoxanthin has attracted considerable attention due to a plethora of its health-promoting properties. Fucoxanthin-producing microalgae exhibit higher growth and fucoxanthin content than macroalgae, suggesting they surrogate macroalgae as the commercial fucoxanthin source. *Tischrysis lutea* exhibited the highest fucoxanthin content (7.94% DW) currently. Despite this, the studies on fucoxanthin production from freshwater microalgae remain scanty. The overall significance of this study is it serves as an antecedent by providing insights into the fucoxanthin-producing microalgae response to different culture factors via a systematic analysis. Although fucoxanthin productivity is highly species-dependent, a consensus on the culture condition of fucoxanthin production was attained among the researchers. For instance, a light intensity ranging from 10 to 100 µmol/m^2^/s could increase the fucoxanthin content of the microalgae. However, the optimal light condition in producing fucoxanthin is species-specific. A low light intensity could enhance the fucoxanthin content, but it might reduce the biomass productivity. The maximum fucoxanthin productivity is often achieved under the culture conditions, where the optimal trade-off between the biomass productivity and fucoxanthin biosynthesis was obtained. Studies on the temperature, pH and salinity factors are required to further fortify this area as the responses of fucoxanthin synthesis to these factors have been rarely discussed. In addition, integrated studies on the factors involved in the biomass and fucoxanthin production are also needed. In-depth information related to the biomass and fucoxanthin production, such as molecular and metabolic studies, is still understudied. To establish sustainable commercial fucoxanthin production, future work should focus on the molecular and metabolic processes of fucoxanthin under these factors, upscaling of fucoxanthin production and economic and environmental assessments of fucoxanthin production (i.e., technoeconomic and life cycle assessment studies).

## Figures and Tables

**Figure 1 marinedrugs-20-00592-f001:**
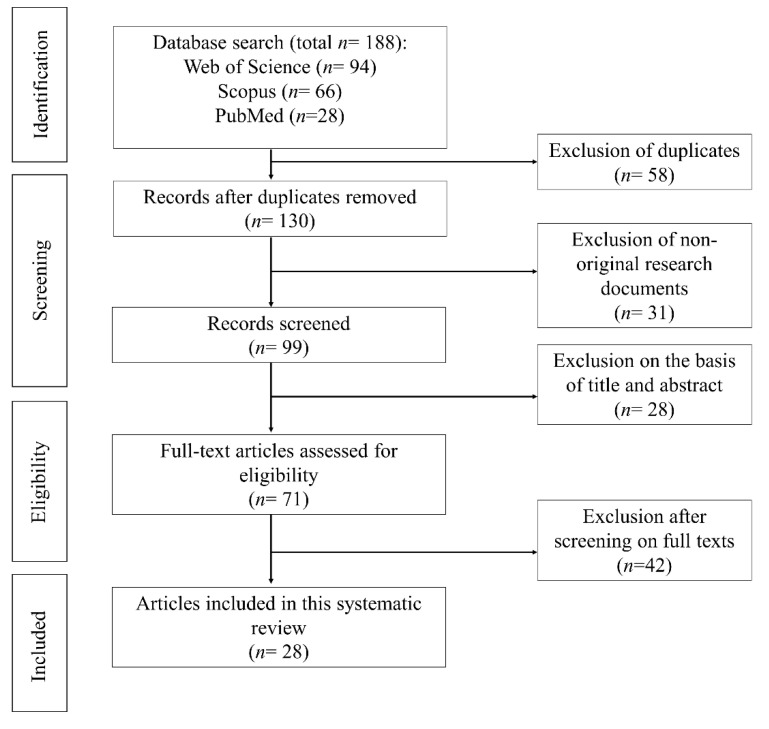
Flow diagram of assessing the articles for the current systematic review.

**Figure 2 marinedrugs-20-00592-f002:**
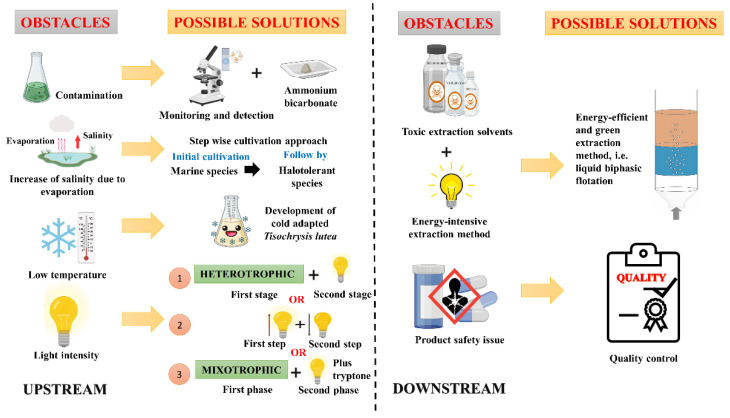
The obstacles in fucoxanthin production and their possible solutions.

**Figure 3 marinedrugs-20-00592-f003:**
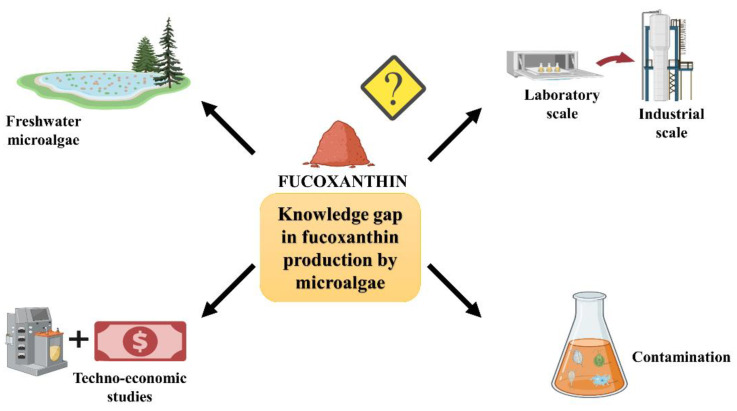
The knowledge gap in the fucoxanthin production by microalgae.

**Table 1 marinedrugs-20-00592-t001:** Previous studies on the responses in terms of the biomass concentration (bc) and fucoxanthin (fx) production of microalgae towards various culture factors.

Microalgae	Type	Factor	Tested Parameter	Highest Bc (g/L)	Optimised Condition for BC	Highest Fx Content (% DW)	Optimised Condition for Fx Content	Reference
*Mallomonas* sp. SBV13	F	Light	10–226 µmol/m^2^/s	3.75	226 µmol/m^2^/s	≈2.50	24 µmol/m^2^/s	Petrushkina et al. [12]
*Isochrysis* sp. CCMP1324	M	Light	30, 60, 120 µmol/m^2^/s	≈2.6	60 µmol/m^2^/s	≈1.30	30 µmol/m^2^/s	Sun et al. [13]
		Nitrogen	25, 50, 100 mg/L	≈2.75	100 mg/L	≈1.35	100 mg/L	
		Phosphorus	1.13, 2.25, 4.50 mg/L	≈2.75	4.50 mg/L	≈1.35	4.50 mg/L	
*Tisochrysis lutea*	M	Light	50, 150, 300, 500 µmol/m^2^/s	1.91	300 µmol/m^2^/s	0.52	50 µmol/m^2^/s	Gao et al. [14]
		Temperature	16.5, 20, 25, 30 °C	1.81	30 °C	≈0.21	25 °C	
*Phaeodactylum tricornutum*	M	Light	100, 150, 210 µmol/m^2^/s	0.29	150 µmol/m^2^/s	4.28	100 µmol/m^2^/s	McClure et al. [15]
		Culture media	f/2, 10 × f/2, f/2 + 10 × nitrate	0.59	10 × f/2	5.92	f/2 + 10 × nitrate	
		Carbon dioxide	0, 1 and 2%	0.45	0%	2.32	1%	
*P. tricornutum*	M	Light	30, 70, 120, 180 µmol/m^2^/s	1.56	70 µmol/m^2^/s	0.75	30 µmol/m^2^/s	Wang et al. [4]
		Nitrogen	0, 75, 150, 300 mg/L	1.61	300 mg/L	0.54	300 mg/L	
		Salinity	5, 10, 20, 30 ‰	1.49	20‰	0.74	20‰	
*Cylindrotheca fusiformis*	M	Light	30, 70, 120, 180 µmol/m^2^/s	1.38	120 µmol/m^2^/s	0.65	30 µmol/m^2^/s	
		Nitrogen	0, 75, 150, 300 mg/L	1.52	300 mg/L	0.61	300 mg/L	
		Salinity	5, 10, 20, 30‰	1.64	30‰	0.58	10‰	
*Cyclotella cryptica* CCMP333	M	Light	10, 20, 30, 40 µmol/m^2^/s	≈1.25	30 µmol/m^2^/s	1.08	10 µmol/m^2^/s	Guo et al. [16]
		Combined	Presence of light and nitrate	1.72	Light and nitrate	1.29	Light and nitrate	
*Isochrysis zhangjiangensis*	M	Light	40, 80, 120, 180, 300 µmol/m^2^/s	2.4	300 µmol/m^2^/s	2.33	40 µmol/m^2^/s	Li et al. [17]
		Carbon dioxide	0, 2, 5%	1.35	5%	2.32	5%	
*Nitzschia laevis*	M	Light	0, 10, 20, 30, 40, 60 µmol/m^2^/s	2.22	0 µmol/m^2^/s	1.11	10 µmol/m^2^/s	Lu et al. [18]
	M	Light	BL: WL (0:1, 1:1, 1:0)	NA	NA	1.20	BL:WL,1:1	
*T. lutea*	M	Light	BL, RL, GL, BL + RL, BL + GL, BL + RL + GL	0.38	BL + RL + GL	1.68	BL + GL	Gao et al. [19]
*Odontella aurita*	M	Light	RL, BL, WL	3.87	RL	1.52	RL	Zhang et al. [20]
		Light	RL:BL (1:9, 2:8, 3:7, 4:6, 5:5, 6:4, 7:3, 8:2, 9:1)	5.65	RL:BL, 8:2	1.62	RL:BL, 8:2	
*Isochrysis galbana*	M	Nitrogen	2, 4, 8, 12 mM	^a^ 7 × 10^6^	4 mM	1.81	4 mM	Nadushan and Hosseinzade [21]
	M	Salinity	25, 35‰	≈^a^ 5.9 × 10^6^	35‰	1.20	35‰	
*Thalassiosira weissflogii*	M	Nitrogen	0, 37.5., 75, 150, 300 mg/L	^a^ 3.5 × 10^6^	75 mg/L	≈0.70	300 mg/L	Marella and Tiwari [5]
		Phosphorus	0, 2.5, 5, 10, 20 mg/L	^a^ 3.6 × 10^6^	10 mg/L	≈0.64	10 mg/L	
		Silicate	0, 35, 70, 140, 280 mg/L	≈^a^ 3.9 × 10^6^	280 mg/L	0.75	280 mg/L	
		Combined	Light (100,300 µmol/m^2^/s) and light (BL, RL, WL)	^a^ 9.5 × 10^6^	300 µmol/m^2^/s and BL	≈1.00	100 µmol/m^2^/s and BL	
*O. aurita*	M	Nitrogen	ILN, IHN, SN	5.84	IHN	2.33	SN	Xia et al. [22]
*Navicula laevis*	M	Silicate	0, 64, 240, 480, 960 mg/L	2.41	480 mg/L	≈1.40	240 mg/L	Mao et al. [23]
*Pavlova* sp. OPMS 30543	M	Nitrogen	sodium nitrate, potassium nitrate, urea, ammonium chloride	1.8	Potassium nitrate	1.27	Sodium nitrate	Kanamoto et al. [24]
		Carbon	glucose, methanol, sodium acetate, sodium bicarbonate	1.79	Sodium acetate	0.73	Methanol	
		Culture media	2X Daigo IMK, f/2, Walne	0.92	2X Daigo IMK	0.26	2X Daigo IMK	
*P. tricornutum*	M	Culture media	Cell-Hi F2P, JWP, WP, FloraMicroBloom, f/2	0.45	Cell-Hi F2P	1.33	Cell-Hi F2P	Butler et al. [25]
*Chaetoceros gracilis*	M	Nitrogen	Urea, sewage water, liquid fertiliser	NA	NA	1.95 mg/L	Urea	Tokushima et al. [26]
		Culture media	Daigo IMK, f/2	^a^ 5.22 × 10^6^	Daigo IMK	2.2 mg/L	Daigo IMK	
*P. tricornutum*	M	Nitrogen	0.5 g/L YE, 1 g/L Try, 0.5 g/L YE + 1 g/L Try	≈0.24	0.5 g/L YE + 1 g/L Try	≈7.5 mg/L	0.5 g/L YE + 1 g/L Try	Hao et al. [27]
*Sellaphora minima*	F	Culture media	FDMed, Guillard and Lorenzen’s WC, Modified COMBO	1.71	FDMed	0.75	FDMed	Gerin et al. [28]
*Nitzchia palea*	F			1.19	FDMed	0.55	FDMed	
*Amphora capitellata*	M	Oxidative stress	0.1 mM hydrogen peroxide + 0.1 mM Fe^2+^, 0.1 mM sodium hypochlorite + 0.1 mM Fe^2+^, 0.1 mM hydrogen peroxide + 0.1 mM sodium hypochlorite	≈0.64	0.1 mM hydrogen peroxide + 0.1 mM sodium hypochlorite	4.18	0.1 mM hydrogen peroxide + 0.1 mM sodium hypochlorite	Erdogan et al. [29]
*T. lutea*	M	Salinity	25–45 g/L	^a^ 4.34 × 10^8^	36.27 g/L	7.94	36.27 g/L	Mohamadnia et al. [30]
		Nitrate	0–0.300 g/L	^a^ 4.34 × 10^8^	0.16 g/L	7.94	0.16 g/L	
		Glucose	0.50–6.50 g/L	^a^ 4.34 × 10^8^	3.90 g/L	7.94	3.90 g/L	
*T. lutea*	M	Temperature	19–35 °C	NA	NA	0.09	25 °C	Beuzenberg et al. [31]
		Light	40–1000 µmol/m^2^/s	NA	NA	0.09	76 µmol/m^2^/s	
		pH	6.7–8.5	NA	NA	0.09	7.4	
*O. aurita*	M	Combined	Light (100, 300 µmol/m^2^/s) and nitrate (6, 18 mM)	6.36	300 µmol/m^2^/s and 18 mM	2.08	100 µmol/m^2^/s and 18 mM	Xia et al. [32]
*T. lutea*	M	Combined	Light (50, 100, 150 µmol/m^2^/s) and nitrate (882, 2646 µM)	0.60	150 µmol/m^2^/s and 2646 µM	1.51	50 µmol/m^2^/s and 2646 µM	Premaratne et al. [33]
*P. tricornutum*	M	Combined	Light (128, 204, 255 µmol/m^2^/s) and silicate (0.3, 3.0 mM)	NA	NA	≈0.75	204 µmol/m^2^/s and 0.3 mM	Yi et al. [34]
*Stauroneis* sp.	M	Combined	Light (WL, BL, RL, GL) and nutrient (f/2 without silicate, f/2 without nitrate, normal seawater, f/2)	NA	NA	0.59	BL and f/2	Parkes et al. [35]
*Chrysotila carterae*	M	Salinity	35–125‰	NA	NA	0.10	35‰	Ishika et al. [36]
*Chaetoceros muelleri*	M	Salinity	35–125‰	NA	NA	0.29	45‰	
*P. tricornutum*	M	Salinity	35–125‰	NA	NA	0.19	45‰	
*T. lutea*	M	Salinity	35–125‰	NA	NA	0.21	45‰	
*Amphora* sp.	M	Salinity	35–125‰	NA	NA	0.15	75‰	
*Navicula* sp.	M	Salinity	35–125‰	NA	NA	0.12	85‰	
*P. tricornutum*	M	Light	10, 20, 30, 40, 50, 100, 150, 200 µmol/m^2^/s	4.80	20 µmol/m^2^/s	1.60	20 µmol/m^2^/s	Yang and Wei [37]
		Light	RL:BL (0:1, 6:1, 1:1, 1:2, 1:0)	5.53	RL:BL (6:1)	1.63	RL:BL (0:1)	

M: marine; F: freshwater; BL: blue light; RL: red light; GL: green light; WL: white light; ILN: initial low nitrogen (6 mM); IHN: initial high nitrogen (18 mM); SN: initial high nitrogen plus supplementary nitrogen (18 mM nitrogen concentration was supplied at the beginning, and an additional 6 mM nitrogen concentration was added on day 3, day 7 and day 11); YE: yeast extract; Try: tryptone; ≈: approximately; ^a^: in cells/mL unit and NA: not available.

**Table 2 marinedrugs-20-00592-t002:** Fucoxanthin contents of several microalgae without an optimal culture condition.

Microalgae	Fucoxanthin Content (% DW)	Reference
*Chaetoceros calcitrans*	0.51	Foo et al. [39]
*Isochrysis galbana*	0.22	
*Skeletonema costatum*	0.04	
*Odontella sinensis*	0.12	
*Phaeodactylum tricornutum*	0.01	
*P. tricornutum*	1.57	Kim et al. [40]
*P.tricornutum*	0.86	Kim et al. [41]
*C. gracilis*	0.22	
*I. galbana*	0.60	
*Isochrysis* aff. *galbana*	1.82	
*Nitzschia* sp.	0.49	
*Cylindrotheca closterium*	0.52	Pasquet et al. [42]
*P. tricornutum*	1.86	Derwenskus et al. [43]
*I. galbana*	0.63	Medina et al. [44]
*Chaetoceros calcitrans*	1.61	Khoo et al. [45]
*C. calcitrans*	1.75	Khoo et al. [46]
*N. laevis*	0.17	Sun et al. [47]

DW: dry weight.

## Data Availability

Not applicable.

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
