# Peer review of "Fucoxanthin Production of Microalgae under Different Culture Factors: A Systematic Review"

_marinedrugs, 2022, doi:10.3390/md20100592_

Round 1
Reviewer 1 Report
This is a very useful and well written review of practical ways to produce fucoxanthin at a large scale with a more cost-effective approach.
Authors: Insert in your review and define; ...diatoms (a single-celled alga which has a cell wall of silica). This would provide useful info for the general public.
Reviewer 2 Report
Introduction
Line 47-49
You wrote: “Fucoxanthin concentration of diatoms was elevated during the exponential phase and accounted for 95% of the total carotenoid. However, the concentration declined in the stationary phase [4, 5]”.
The reference number 4 said: “Growth and accumulation of fucoxanthin and EPA of diatoms may vary considerably across species and/or strains and are dependent on culture conditions such as nutrient concentrations and environmental factors”
This comment is general, it didn’t take into consideration the global literature on diatoms. In this paper, it was based on only two publications and it concerns only 3 microalgae: Thalassiosira weissflogii, Phaeodactylum tricornutum and Cylindrotheca fusiformis.
Moreover, you said fucoxanthin accounted for 95% of the total carotenoids: all the carotenoids were not quantified in the two publications cited, how did you find this number 95%?
Lines 61-64 : a review was published in march 2022 in marine drugs. it answered to your questions. I saw that it was published after your bibliographic search.
Results :
I checked on google scholar “Fucoxanthin” AND “microalgae” there is 10 500 results. I check then the same keywords in web of science and I found 308 results. I think than some research have been forgotten. The flow diagram is methodical but, it did not take into account the quality and the effective content of the publications.
Discussion
Table 2 : lack of references
Line 120-123 : “However, culturing marine microalgae in the photobioreactors could produce problems associated with corrosion and crystallization”. If you have read the literature it’s only in stainless steel fermenters.
Line 147 : dark condition: the culture were in mixotrophy condition?
Line 169-172 : confusion with the effect of light in autotrophy and the heterotrophic culture condition.
Line 262-286 : It was not possible to draw a conclusion, the concentration of nitrogen are cited for different algae, system of culture source of nitrogen. Nothing was comparable.
The paragraph with nitrogen and culture media was redundant. All the culture media contains nitrogen.
Figure 2 : if you had an efficient contamination, detection, monitoring and controlling were not a possible solution. It did not eliminate the contaminant…
Figure 3 :
From an environmental point of view, given the demand for fresh water for human consumption, it is inappropriate to suggest growing algae in fresh water to make fucoxanthin when seawater covers 70% of our planet and does not necessarily require the addition of nutrients for algae growth.
From an ecological point of view, this publication did not identify the best fucoxanthin-producing microalgal species. Most of these species are from marine origin.
There are too much misunderstanding of the litterature, I reject the publication.
Reviewer 3 Report
1. Abstract: Fucoxanthin productivity is also heavily dependent on the microalgal species. Lines 28-30 describes the proposed cultivation parameters to maximize fucoxanthin productivity – yet, the optimal conditions are highly-species dependent. Lines 34-35: Did the authors perform an economic analysis to come into this conclusion? If so, it should be mentioned in the abstract. At a glance, the abstract seems vague and raises questions regarding the exact methodology followed to derive the proposed culture condition.
2. The authors should improve the grammar/technical writing of the manuscript – ex:
“…properties such as anti-diabetes, antiobesity, anticancer and antioxidant”. It is advised to thoroughly proofread the text and adjust the language so that it is more technically sound. Moreover, some phrases are not technical and not befitting a scientific manuscript – ex: line 68 – “practical ways”, line 71 – “most of them”. Lines 94-95: “lies” is not appropriate given the context.Authors should go through the entire manuscript and improve the text substantially.
3. There are some sentences which have no weight and lack context – ex: “Fucoxanthin concentration of diatoms was elevated during the exponential phase and accounted for 95% of the total carotenoid.” – this is extremely vague and hence add little value – what was the species of the diatom? How does this sentence fit with rest of the text? “Tisochrysis lutea is also known as I. galbana T-Iso [13].” – another random sentence which does not fit with the preceding sentences. Authors are advised to adjust the flow of the entire manuscript.
4. The authors should clearly highlight the novelty of their review in the introduction.
5. Lines 56 & 57 are contradictory – it is unclear what the authors are trying to convey to the reader.
6. Line 76-77: “(1) inclusion of studies, specifically fucoxanthin production of microalgae (2) exclusion of studies involv-78 ing environmental studies.” – lacks clarity. The reader would not be able to comprehend what these criteria are.
7. Lines 106-116 – this seems more fitting in the introduction section of the manuscript. Also it disrupts the flow of the discussion.
8. Lines 123-124 – it is well known that fucoxanthin is produced by a large number of marine microalgae species. The relevance of reporting this is unclear – concluding this does not require a comprehensive analysis, as it is apparent. This seems more fitting for the introduction. The authors should improve the discussion futher.
9. Figure 2 & 3 – should be represented more technically
10. The discussion on culture parameters is quite lengthy, and should be substantially reduced. The effect of culture parameters is species dependent, and are relatively well known – they have been discussed in previous reviews. In contrast, section 4 and 5 should be further elaborated – as they are the most interesting sections of the review. Nevertheless, these sections are very short and does not provide a holistic view of the current status. Section 4 is just 3 paragraphs. In section 5, the authors have not elaborated on the identified challenges in detail, and have missed certain major challenges including process economics, the requirement to develop robust downstream processes, concerns regarding product safety, etc. Similarly, section 6 could also be more comprehensive. The authors should also consider changing the title to reflect these changes.
11. Lines 710-714 – as mentioned previously, proposing such conditions is meaningless as it is heavily species dependent.
Round 2
Reviewer 3 Report
1. The authors have repeatedly mentioned that their study’s novelty derives from the comprehensive analysis of culture conditions that affect fucoxanthin production. Yet, the following reviews have also described them in detail –
https://doi.org/10.1016/j.biortech.2021.126170
https://doi.org/10.1016/j.biotechadv.2021.107865
The authors’ assessment on the novelty of their study is therefore, lacking.
2. I strongly disagree with presenting generalized culture conditions as “optimal” for fucoxanthin production, when the optimal conditions to maximize productivity are species-specific. Such absolute statements should be removed from the manuscript. Instead, the authors can suggest that certain conditions (ex: low light) can induce the synthesis of fucoxanthin, while explicitly remarking that optimal conditions are species-specific. Moreover, it is noteworthy that the low-light condition suggested by the authors can induce the biosynthesis of fucoxanthin in cells, it might also reduce biomass productivity. The maximum fucoxanthin productivity is often achieved under conditions that achieve the optimal tradeoff between biomass productivity and fucoxanthin biosynthesis.
3. Authors have mentioned that they have made amendments in response to comment #2 from the previous round of peer review. However, this has not been reflected in the text.
4. Authors’ reasoning behind the response to comment #9 from the previous round of peer review, is understandable. However, the manuscript can be reframed so that the point they are trying to make is highlighted (i.e. freshwater microalgae can also produce fucoxanthin), without including obvious statements. Most notably, the authors do not need to include both of these sentences (one concise and to-the-point sentence would suffice). - “Firstly, most fucoxanthin production studies pay atten-655 tion to marine species. Marine microalgae have been described as potential biomass pro-656 ducers and targeted as fucoxanthin sources under different conditions (Table 1).”.
5. There are still several typographical errors in the manuscript.
6. Increase the font sizes of figure 2 and 3 to improve the clarity. In figure 2, some of the “issues” have not been labeled.
7. The sentences in lines 496-497 seem out of place (an abrupt change of pace). This is due to the newly added text preceding these sentences.
